# A Novel Waveform Optimization Method for Orthogonal-Frequency Multiple-Input Multiple-Output Radar Based on Dual-Channel Neural Networks

**DOI:** 10.3390/s24175471

**Published:** 2024-08-23

**Authors:** Meng Xia, Wenrong Gong, Lichao Yang

**Affiliations:** 1School of Communication and Information Engineering, Xi’an University of Science and Technology, Xi’an 710054, China; xia228@163.com (M.X.); gongwr@163.com (W.G.); 2Science and Technology on Communication Information Security Control Laboratory, Jiaxing 314033, China

**Keywords:** MIMO radar, orthogonal frequency, OFDM-LFM, waveform design, dual-channel neural networks

## Abstract

The orthogonal frequency-division multiplexing (OFDM) mode with a linear frequency modulation (LFM) signal as the baseband waveform has been widely studied and applied in multiple-input multiple-output (MIMO) radar systems. However, its high sidelobe levels after pulse compression affect the target detection of radar systems. For this paper, theoretical analysis was performed, to investigate the causes of high sidelobe levels in OFDM-LFM waveforms, and a novel waveform optimization design method based on deep neural networks is proposed. This method utilizes the classic ResNeXt network to construct dual-channel neural networks, and a new loss function is employed to design the phase and bandwidth of the OFDM-LFM waveforms. Meanwhile, the optimization factor is exploited, to address the optimization problem of the peak sidelobe levels (PSLs) and integral sidelobe levels (ISLs). Our numerical results verified the correctness of the theoretical analysis and the effectiveness of the proposed method. The designed OFDM-LFM waveforms exhibited outstanding performance in pulse compression and improved the detection performance of the radar.

## 1. Introduction

Multiple-input multiple-output (MIMO) radar has exhibited exceptional performance in target detection, parameter estimation, radio frequency stealth, and anti-jamming owing to its transmitted waveform diversity [1,2,3,4,5]. MIMO radar achieves better target detection, parameter estimation, and anti-jamming capabilities by emitting different waveforms. Thus, the waveform design of MIMO radar is one of the key technologies of the system. It utilizes optimization theory to design waveforms that meet radar performance requirements under different tasks, environments, and resource constraints [6,7,8]. Extensive research on the waveform design for MIMO radar systems has been performed for nearly two decades [9,10,11,12,13], primarily focusing on the peak sidelobe levels (PSLs), integrated sidelobe levels (ISLs), and cross-correlation properties of the waveform. In recent years, as the research has progressed, many new findings and applications have been introduced. In [14], the problem of waveform design in coherent MIMO radar was studied and two new methods for constant modulus waveform design were introduced, aimed at reducing PSLs. By introducing auxiliary variables, the corresponding non-convex optimization problem can be addressed and a sufficiently low PSL can be obtained under the constant modulus constraint. A method for adaptive waveform design using multitone sinusoidal frequency modulation was also introduced [15]. It expresses the waveform parameters in terms of Fourier coefficients and it constrains the frequency spectrum, autocorrelation, and ambiguity function shapes of the waveform by modifying these design parameters. The resulting optimized design considerably improved the PSL in a specified region of the range-Doppler plane without increasing the time-bandwidth product. The waveform design problem in MIMO radar systems for target detection was studied [16], and a new algorithm based on maximization–minimization was proposed. A simple quadratic function was formulated to minimize the objective function, which reduced the computational complexity of the new algorithm compared with similar algorithms. The joint design problem of the transmission waveform and the target detector in radar systems was also discussed [17]. Based on end-to-end training of radar systems, two new learning-based waveform and detector design methods were proposed. The designed waveforms exhibited good environmental adaptability and met the design constraints. A comprehensive method for MIMO radar waveform design was proposed [18], which shaped the beam patterns in MIMO radar systems under constraints while ensuring waveform monotonicity, the required spectrum utilization, and orthogonality. An efficient iterative method was also proposed, to address this non-convex optimization problem.

A linear frequency modulation (LFM) signal can address the contradiction between radar range and resolution while also offering the advantage of being insensitive to Doppler frequency. Therefore, the application of the orthogonal frequency-division multiplexing linear frequency modulation (OFDM-LFM) waveform in MIMO radar systems has been extensively studied. By analyzing waveform sidelobe performance, a joint optimization design method for OFDM chirp signals was proposed that used genetic algorithms and sequential quadratic programming [19]. A set of correlated LFM waveforms was developed to address the challenge of generating constant-modulus waveforms in a MIMO radar system [20]. The method was used to analyze the covariance matrix of the LFM waveform set, from which the transmitted beam pattern was derived. Finally, a constrained nonlinear optimization method was employed to solve the problem, and the chirp waveform set was obtained. The designed waveform had the properties of a constant envelope and could be easily generated to accurately match the desired transmitted beam pattern. Owing to its good correlation and large time-bandwidth product characteristics, the OFDM waveform was used in multi-static radar systems [21], and the ambiguity function was used for analyzing the target resolution capability of the waveform. As the performance of the OFDM-LFM signal degrades during the estimation of the direction of arrival (DOA), a novel OFDM-LFM waveform design method based on the discrete Fresnel transform scale was proposed [22]. This method can efficiently generate orthogonal LFM signals with an unlimited number of subcarriers and can be rapidly solved using the fast Fourier transform (FFT). As the elevation pulse internal beam control methods can considerably enhance the transmission efficiency and multitasking capabilities of MIMO radar systems, a novel radar system that integrated elevation pulse internal beam control with OFDM chirp signals was proposed [23]. This system notably improved the radar performance and its resilience to interference signals. Moreover, a method for jointly designing the OFDM-LFM waveform and the receiving filter for spatially heterogeneous clutter was introduced [24]. A novel algorithm based on iterative sequence optimization was also presented for the direct optimization of multiple sets of sub-channel durations within the OFDM-LFM waveform, which ultimately enhanced the output signal-to-noise ratio.

Deep learning, represented by convolutional neural networks (CNNs), was initially applied to address problems with SAR images in radar signal processing [25,26,27,28,29]. In recent years, the application of CNNs to other areas of radar signal processing has also been extensively studied. A novel algorithm for counting the number of individuals was introduced, using CNNs in low-radiation impulse radio ultra-wide bandwidth (IR-UWB) radar [30]. Using this approach, multi-scale range-time maps were extracted from the received data, and CNNs were used to further categorize the number of individuals. Finally, the superiority of this algorithm was experimentally verified. For ship detection in radar images using deep learning, two ship detectors based on faster region CNNs (R-CNNs) were proposed [31], which exhibited high performance even in complex sea conditions with dense multi-target scenes. For detecting small, slow, and low-speed targets, an innovative approach was proposed for radar feature extraction in a slow-time dimension, using one-dimensional CNNs [32]. Thus, the complexities associated with frequency and wavelet transforms were avoided. The proposed approach not only simplified the analysis but also demonstrated excellent classification capabilities, yielding promising results. The application of deep learning to radar waveform optimization has also attracted research attention. A comprehensive waveform design method to minimize the weighted sum of almost entirely metrics under a constant-modulus constraint was proposed, and a deep learning framework was derived to handle this problem [33]. A data-driven robust DOA estimation framework was proposed for MIMO radar via deep neural networks [34]. MIMO radar unimodular waveform design was transformed into an unconstrained minimization problem that could be addressed using the Riemannian gradient descent algorithm, and a deep unfolding network was proposed to address the problem [35].

Considering the powerful information representation capabilities of CNNs, in this paper we introduce them into MIMO radar systems for optimizing OFDM-LFM waveforms. Unlike the existing single-network structured deep learning waveform optimization methods, a dual-channel-CNNs method based on ResNeXt is proposed, to optimize OFDM-LFM waveforms by deeply analyzing the causes of high sidelobe levels in OFDM-LFM waveforms. The phase optimization channel draws on the idea of ResNeXt grouping convolution to achieve this, whereas the bandwidth optimization channel adopts a classic convolutional neural network (CNN) structure. The phase and bandwidth parameters of the OFDM-LFM waveforms are simultaneously optimized using this structure. A new loss function is simultaneously constructed that can be adjusted between the PSL and the ISL based on the optimization factor. The optimized OFDM-LFM waveforms have lower sidelobe levels and eliminate the periodicity of high sidelobe levels. This considerably enhances the target detection capability. The innovations of this study are as follows:
The causes of the periodic high sidelobes in the OFDM-LFM waveform applied in MIMO radar systems were analyzed in detail, which revealed that the periodicity is related to pulse width.In order to design OFDM-LFM waveforms with good correlation properties, a ResNeXt-based dual-channel CNNs method is presented, to optimize the phase and bandwidth of the OFDM-LFM waveforms. Meanwhile, a new adjustable objective function is proposed, which can optimize both the PSL and the ISL simultaneously by utilizing the optimization factor.Extensive numerical simulations were employed to validate the performance of the proposed CNNs method for OFDM-LFM waveform design. The experimental results show that the designed waveforms have better target detection performance compared to the traditional OFDM-LFM waveforms.

The rest of this paper is organized as follows. The MIMO radar signal model and the sidelobe properties of the OFDM-LFM are analyzed in Section 2. Thereafter, to design OFDM-LFM waveforms with good correlation properties, dual-channel CNNs and two different loss functions are presented in Section 3. In Section 4, the simulations and data analysis are discussed, to verify the performance of the proposed method. Finally, our conclusions are obtained in Section 5.

## 2. MIMO Radar Signal Model

### 2.1. OFDM-LFM Signal

Consider a coherent MIMO radar system with *N* transmitting antennas, where the antennas transmit different baseband waveforms with orthogonal carrier frequencies. When the adopted baseband waveform is a linear frequency modulation signal, it is known as an OFDM-LFM waveform. For uniform linear arrays with a spacing of *d*, the *n*th transmitting signal can be expressed as follows:
(1)$$s_{n}\left(t\right)=a\left(t\right)\exp\left(j\pi kt^{2}\right)\exp\left(j2\pi f_{n}t\right)$$
where $a\left(t\right)=\left\{\begin{array}{c} 1,\;\;\left|t\right|\leq T\\ 0\;\;else \end{array}\right.$ is a rectangular pulse with duration *T*, and where $f_{n}=f_{0}+\left(n-\frac{N+1}{2}\right)\Delta f$ is the carrier frequency of the *n*th antenna, $f_{0}$ denotes the center frequency, Δf indicates the transmission frequency interval, and *k* is the modulation frequency of the LFM signal. By ignoring the influence of the target distance, the synthesized signal of the transmitted signal from a target with direction θ can be written as
(2)$$p\left(t\right)=\sum_{n=1}^{N}a\left(t\right)\exp\left(j\pi kt^{2}\right)\exp\left(j2\pi f_{n}t\right)\exp\left(j\xi_{n}\right)=a_{t}^{T}\left(\theta \right)S$$
where $\xi_{n}$ is the phase induced by the transmit array structure, $a_{t}\left(\theta \right)={\left[1,\ldots ,e^{j\xi_{n}},\ldots ,e^{j\xi_{N}}\right]}^{T}$ denotes the transmit array steering vector, and $S={\left[s_{1},\ldots ,s_{n},\ldots ,s_{N}\right]}^{T}$. To ensure that the signals transmitted by each antenna are mutually orthogonal, $T=\frac{W}{\Delta f}$ with *W* is an integer. When the signals emitted from each array element are orthogonal, the superimposed signals do not produce a directional pattern. Moreover, the radiated energy is equal throughout the entire space. At the receiving end, the echo signal received by the *m*th antenna can be expressed as follows:
(3)$$x_{m}\left(t\right)=p\left(t\right)\exp\left(j\gamma_{m}\right)+n_{m}\left(t\right)=a_{t}^{T}\left(\theta \right)S\exp\left(j\gamma_{m}\right)+n_{m}\left(t\right)$$

For uniform linear arrays, $\gamma_{m}=-\frac{2\pi d(m-1)}{\lambda }\sin\left(\phi \right)$, $a_{r}\left(\phi \right)={\left[1,\ldots ,e^{j\gamma_{m}},\ldots ,e^{j\gamma_{M}}\right]}^{T}$ is the array steering vector and $n_{m}\left(t\right)$ is a Gaussian white noise with zero mean and variance one. The received echo signal vector of the entire array is
(4)$$X\left(t\right)={\left[x_{1}\left(t\right),x_{2}\left(t\right),\ldots ,x_{M}\left(t\right)\right]}^{T}=a_{r}\left(\phi \right)a_{t}^{T}\left(\theta \right)S+n\left(t\right)$$
where $n\left(t\right)={\left[n_{1}\left(t\right),\;n_{2}\left(t\right),\;\ldots \;,\;n_{M}\left(t\right)\right]}^{T}$ is the received noise vector.

At the receiving end, the MIMO radar system performs receiving beamforming on the echo signal. This involves compensating for the phase of the received signals from each antenna before superimposing them, which yields the following results:
(5)$$y_{beam}\left(t\right)=a_{r}^{H}\left(\phi \right)a_{r}\left(\phi \right)a_{t}^{T}\left(\theta \right)S+a_{r}^{H}\left(\phi \right)n\left(t\right)$$
the signals are matched and separated from *N* transmitted signals $s_{n}\left(t\right)$ after receiving beamforming, which is also a characteristic of the MIMO radar signal processing. To achieve complete signal separation, the carriers of transmitted signals must be orthogonal to each other. The orthogonality of the transmitted signal carrier waves was analyzed for this paper. The analysis was performed on the carriers of the transmitted signals $s_{n}\left(t\right)$ and $s_{m}\left(t\right)$, without considering the effect of the baseband waveform, and their cross-correlation results can be defined as follows:
(6)$$\begin{array}{c} R=\int_{-\infty }^{+\infty }a\left(t\right)\exp\left(j2\pi f_{n}t\right)a\left(t-\tau \right)\exp\left[-j2\pi f_{m}\left(t-\tau \right)\right]dt\\ \;\;\;\;=\frac{\sin[\pi (n-m)\Delta fT]}{\pi (n-m)\Delta f}\exp\left[j2\pi \left(f_{0}+\left(m-\frac{N+1}{2}\right)\Delta f\right)\tau \right] \end{array}$$
in the above formula, only when ΔfT is an integer—i.e., T=W/Δf, sin[π(n−m)ΔfT] equals zero—are the signals transmitted by each antenna orthogonal to one other and independent of the baseband waveform.

The matching output of the received beamforming result $y_{beam}\left(t\right)$ with the *n*th transmitted signal $s_{n}\left(t\right)$ can be expressed as follows:
(7)$$\begin{array}{c} y_{n}\left(t\right)=\int_{-\infty }^{+\infty }y_{beam}\left(t\right)s_{n}^{*}\left(t-\tau \right)dt\\ \;\;\;\;\;\;\;\;\;=c_{0}a_{r}^{H}\left(\phi \right)a_{r}\left(\phi \right)\exp\left(j\xi_{n}\right)+\int_{-\infty }^{+\infty }a_{r}^{H}\left(\phi \right)n\left(t\right)s_{n}^{*}\left(t-\tau \right)dt \end{array}$$
where $c_{0}$ is the baseband waveform matching gain. After performing correlation processing on $y_{beam}\left(t\right)$ and *N* transmit waveforms, a matched filter vector signal $Y\left(t\right)={\left[y_{1}\left(t\right),\ldots ,y_{n}\left(t\right),\ldots ,y_{N}\left(t\right)\right]}^{T}$ can be obtained. This signal also has to undergo equivalent transmit beamforming, which involves compensating for phase $\xi_{n}$ and performing an in-phase addition to obtain the following result:
(8)$$z\left(t\right)=a_{t}^{H}\left(\theta \right)Y\left(t\right)=c_{0}a_{r}^{H}\left(\phi \right)a_{r}\left(\phi \right)a_{t}^{H}\left(\theta \right)a_{t}\left(\theta \right)+n_{z}\left(t\right)$$
where $n_{z}\left(t\right)$ is the noise processing result. The aforementioned process is equivalent to synthesized signal processing, which can be further proved by performing two-dimensional matched filtering on $y_{beam}\left(t\right)$ in both temporal and spatial domains. The corresponding matched filter is
(9)$$h\left(t\right)=\sum_{n=1}^{N}s_{n}^{*}\left(-t\right)\exp\left(-j\xi_{n}\right)$$
which means the processing result is $z\left(t\right)=y_{beam}\left(t\right)⊗h\left(t\right)$. The filter has completed two-dimensional matching processing for $y_{beam}\left(t\right)$ in both temporal and spatial domains, and the processing results are the same as the previously described process of first matching and separation, followed by equivalent transmit beam forming.

### 2.2. Analysis of Sidelobe Performance

The sidelobe properties of MIMO radar with the OFDM-LFM waveform have been comprehensively analyzed using synthesized signal processing, and the optimization method for suppressing high sidelobes is investigated in this paper. As the gain $a_{r}^{H}\left(\phi \right)a_{r}\left(\phi \right)$ of the receive beamforming and the noise does not affect the sidelobe properties of the waveform, the received signal is re-expressed as follows:
(10)$$r\left(t,\tau \right)=\sum_{n=1}^{N}a\left(t-\tau \right)\exp\left[j\pi k{\left(t-\tau \right)}^{2}\right]\exp\left[j2\pi f_{n}\left(t-\tau \right)\right]\exp\left(j\xi_{n}\right)$$
where τ is the time delay induced by the target distance. Applying synthesized signal processing yields [19]
(11)$$\begin{array}{c} Z\left(\tau \right)=\int_{-\infty }^{+\infty }\sum_{m=1}^{N}a\left(t\right)e^{j\pi kt^{2}}e^{j2\pi f_{m}t}e^{j\xi_{m}}\cdot \sum_{n=1}^{N}a\left(t-\tau \right)e^{-j\pi k{\left(t-\tau \right)}^{2}}e^{-j2\pi f_{n}\left(t-\tau \right)}e^{-j\xi_{n}}dt\\ \;\;\;\;\;\;\;\;\;=\int_{-\infty }^{+\infty }a\left(t\right)a\left(t-\tau \right)\sum_{m=1}^{N}e^{j\pi kt^{2}}e^{j2\pi f_{m}t}e^{j\xi_{m}}\cdot \sum_{n=1}^{N}e^{-j\pi k{\left(t-\tau \right)}^{2}}e^{-j2\pi f_{n}\left(t-\tau \right)}e^{-j\xi_{n}}dt\\ \;\;\;\;\;\;\;\;\;=F(\tau )+R(\tau )+L(\tau )\; \end{array}$$
where $F\left(\tau \right)=\sum_{m=1}^{N}e^{j2\pi f_{m}\tau }\frac{\sin\pi k\tau (T-\left|\tau \right|)}{\pi k\tau (T-\left|\tau \right|)}\left(T-\left|\tau \right|\right)$ determines the mainlobe level of the processed result and R(τ) influences the sidelobe properties, which can be expressed as
(12)$$\begin{array}{c} R\left(\tau \right)=\int_{-\infty }^{+\infty }a(t)a(t-\tau )\sum_{m=1}^{N-1}\sum_{n=m+1}^{N}e^{j\pi k{\left[t-(t-\tau )\right]}^{2}}e^{j2\pi (f_{m}-f_{n})t}e^{j2\pi f_{n}\tau }e^{j(\xi_{m}-\xi_{n})}dt\\ \;\;\;\;\;\;\;\;\;=\sum_{m=1}^{N-1}\sum_{n=m+1}^{N}e^{j(\xi_{m}-\xi_{n})}e^{j\pi (f_{m}+f_{n})\tau }\frac{\sin[\pi \left(k\tau +f_{m}-f_{n}\right)\left(T-\left|\tau \right|\right)]}{\pi \left(k\tau +f_{m}-f_{n}\right)\left(T-\left|\tau \right|\right)}\left(T-\left|\tau \right|\right) \end{array}$$
similarly, the following expression can be derived:
(13)$$L\left(\tau \right)=\sum_{m=2}^{N}\sum_{n=1}^{m-1}e^{j(\xi_{m}-\xi_{n})}e^{j\pi (f_{m}+f_{n})\tau }\frac{\sin[\pi \left(k\tau +f_{m}-f_{n}\right)\left(T-\left|\tau \right|\right)]}{\pi \left(k\tau +f_{m}-f_{n}\right)\left(T-\left|\tau \right|\right)}\left(T-\left|\tau \right|\right)$$

Based on the above results, the correlation properties of Z(τ) are analyzed in detail. The mainlobe is determined by F(τ), whereas the sidelobe properties are jointly determined by F(τ), R(τ), and L(τ). Due to the symmetry of R(τ) and L(τ), only F(τ) and R(τ) are analyzed here.

F(τ) can be further represented as follows:
(14)$$\begin{array}{c} F\left(\tau \right)=\sum_{m=1}^{N}e^{j2\pi f_{m}\tau }\frac{\sin\pi k\tau (T-\left|\tau \right|)}{\pi k\tau (T-\left|\tau \right|)}\left(T-\left|\tau \right|\right)\\ \;\;\;\;\;\;\;\;\;=e^{j\pi (N-1)\Delta f\tau }\frac{\sin(\pi N\Delta f\tau )}{\sin(\pi \Delta f\tau )}\frac{\sin\pi k\tau (T-\left|\tau \right|)}{\pi k\tau (T-\left|\tau \right|)}\left(T-\left|\tau \right|\right) \end{array}$$
when τ≠0 the sidelobe is composed of three parts: the main function $\frac{\sin\pi k\tau (T-\left|\tau \right|)}{\pi k\tau (T-\left|\tau \right|)}\left(T-\left|\tau \right|\right)$, the periodic sampling function $\frac{\sin(\pi N\Delta f\tau )}{\sin(\pi \Delta f\tau )}$, and the exponential phase $e^{j\pi (N-1)\Delta f\tau }$. When $\tau =\pm \frac{m}{\Delta f}$ (m=1,2…W) the sampling function attains its maximum value *N* and periodic distance sidelobes appear. Figure 1a shows the main function and periodic sampling function, and Figure 1b shows the periodic distance sidelobe results. The corresponding parameters are *N* = 25, *W* = 8, and Δf=1 MHz.

Based on Equation (Equation 12), R(τ) can be rewritten as
(15)$$R\left(\tau \right)=\sum_{p=-N+1}^{-1}e^{j\xi_{p}}e^{j\pi (N+1)\Delta f\tau }\frac{\sin[\pi \left(k\tau +p\Delta f\right)\left(T-\left|\tau \right|\right)]}{\pi \left(k\tau +p\Delta f\right)\left(T-\left|\tau \right|\right)}\left(T-\left|\tau \right|\right)\frac{\sin[\pi (N+p)\Delta f\tau ]}{\sin(\pi \Delta f\tau )}$$
where p=m−n, detailed derivation can be obtained in Appendix A. R(τ) can be viewed as a weighted sum of the sinc function (referred to as the main function) whose peak position is shifted by pΔf. The weighting coefficient of the main function is $\frac{\sin[\pi (N+p)\Delta f\tau ]}{\sin(\pi \Delta f\tau )}$, and it attains the maximum value at $\tau =\pm \frac{m}{\Delta f}$ (m=0,1…W) while ensuring the presence of high-range sidelobes at these positions. The main function and corresponding sampling function are shown in Figure 2a; R(τ) and L(τ) are shown in Figure 2b.

In summary, when the OFDM-LFM waveforms satisfy frequency orthogonally and are applied to MIMO radar, the echo processing results exhibit periodic high-range sidelobes at $\tau =\pm \frac{m}{\Delta f}$ (m=1,2…W). These high-range sidelobes can interfere with radar target detection, and the sidelobes of strong targets can block the mainlobe of weaker ones. Therefore, in this paper, a novel optimization method based on CNNs is proposed, to design OFDM-LFM waveforms that effectively suppress the high-range sidelobes and enhance the target detection of the echo signal.

## 3. Structure of Neural Networks

### 3.1. Dual-Channel CNNs Model

The CNN is a type of deep learning model that is extensively employed for classification and recognition tasks that involve signal data, such as images, videos, and speech. Its core idea is to extract features via convolutions, pooling operations, and other techniques, to map the input data into a high-dimensional feature space. The features are then classified or regressed through fully connected layers. Deep learning, represented by CNN, has recently shown promising results in fields such as computer vision, natural language processing, signal processing, and remote sensing science. In this paper, CNN is applied to waveform optimization for MIMO radar to generate waveforms with good sidelobe properties. In general, the greater the number of layers in a CNN, the stronger its ability to represent the characteristics of things. However, simply increasing the number of layers in a neural network leads to problems, such as vanishing gradients or exploding gradients. These problems can be solved via normalization. However, as the number of layers in a network significantly increases, there arises a problem of degradation in representation ability. This is because as the neural network becomes deeper the correlation between the back-propagated gradients worsens and the update of the gradients will be meaningless. To address this issue, ResNet, which is based on a residual network, was introduced in [36]. In [37], a parallel stack topology was also employed to replace the original ResNet three-layer convolution module, which improved the model accuracy without considerably increasing the parameter size. Moreover, the consistent topology and reduced hyperparameters make the model easier to transplant. In this paper, we address the issue of periodic sidelobes in the synthesized processing of the LFM signal for orthogonal-frequency MIMO radar. Using the bandwidth and phase of the LFM signal as the optimization parameters, dual-channel CNNs based on the ResNeXt architecture are constructed for designing OFDM-LFM waveforms. The novel neural networks proposed in this paper for OFDM-LFM waveform design are shown in Figure 3, which comprises two channels for phase and bandwidth optimization.

The initialization input for the phase-optimization channel is uniformly distributed data ranging from 0 to 2π, whose size is equal to the number of antennas in the MIMO radar system. The data are first preprocessed via convolution and normalization; they are then passed through two ResNeXt-based residual modules. After passing through a fully connected layer and a new phase activation function, the data enter the sidelobe loss function. The optimized phase value range is [−π,π]. The new phase activation function employed in this paper is expressed as follows:
(16)$$\pi \times \tanh=\pi \frac{1-e^{-2x}}{1+e^{-2x}}$$

For bandwidth optimization, the input to this channel comprises uniformly distributed data ranging from 1 to $N_{B}$, where $N_{B}$ is the bandwidth modulation factor. The data are processed via convolution, normalization, and pooling before being connected to the fully connected layer. Then, they are processed through a bandwidth activation function. The bandwidth of a single transmission signal range is $[1,\;N_{B}]\times \Delta f$, and the new bandwidth activation function is given by
(17)$$\left[\left(N_{B}-1\right)\times \frac{\left(\tanh+1\right)}{2}+1\right]\times \Delta f=\left(\frac{N_{B}-1}{1+e^{-2x}}+1\right)\times \Delta f$$

The OFDM-LFM waveform optimized using the novel neural networks-based method can be written as
(18)$$s_{D}\left(n\right)=a\left(t\right)\exp\left(j\pi k_{n}t^{2}\right)\exp\left(j2\pi f_{n}t\right)\exp\left(\psi_{n}\right)$$
where $k_{n}$ and $\psi_{n}$ are the bandwidth and phase obtained via the neural networks optimization, respectively.

### 3.2. Loss Function

In radar waveform optimization, the PSL and ISL are important indicators often employed to evaluate the performance of waveform target detection. For a radar waveform *s*(t), its aperiodic autocorrelation is represented as $R_{a}\left(\tau \right)=\int_{-\infty }^{+\infty }s\left(t\right)s^{*}\left(t-\tau \right)dt$. The PSL can be defined as
(19)$$PSL=10\log10\frac{\max{\left|R_{a}\left(\tau \right)\right|}^{2}}{{\left|R_{a}\left(0\right)\right|}^{2}},\;\;\tau \neq 0$$
where $\max\left|R_{a}\left(\tau \right)\right|$ is the maximum value of the sidelobe levels and $\left|R_{a}\left(0\right)\right|$ is the autocorrelation mainlobe of the waveform. Therefore, waveform optimization with the PSL as the evaluation criterion involves solving for the minimization of the peak sidelobe levels. Similarly, ISL is defined as
(20)$$ISL=10\log10\frac{\sum_{\tau \neq 0}{\left|R_{a}\left(\tau \right)\right|}^{2}}{{\left|R_{a}\left(0\right)\right|}^{2}}$$
where $\sum_{\tau \neq 0}{\left|R_{a}\left(\tau \right)\right|}^{2}$ denotes the sum of all sidelobe energies, so the ISL-based optimization problem can be interpreted as minimizing the sum of all the autocorrelation sidelobe energies.

The application of the OFDM-LFM waveform to orthogonal-frequency MIMO radar will result in periodic high-range distance sidelobes, which will affect target detection and cause the mainlobe of weak targets to be blocked by the sidelobes of strong targets. Here, when optimizing OFDM-LFM waveforms with dual-channel CNNs, a new optimization objective function is constructed, to simultaneously consider the optimization problems of the PSL and ISL, as follows:
(21)$$f=\varepsilon \frac{\max{\left|R_{a}\left(\tau \right)\right|}^{2}}{{\left|R_{a}\left(0\right)\right|}^{2}}+\left(1-\varepsilon \right)\frac{\sum_{\tau \neq 0}{\left|R_{a}\left(\tau \right)\right|}^{2}}{{\left|R_{a}\left(0\right)\right|}^{2}}$$
where 0≤ε≤1 is the optimization factor, and the corresponding optimization problem proposed in this paper is
(22)$$\begin{array}{c} \mathrm{minimize}[\varepsilon \frac{\max{\left|Z(\tau )\right|}^{2}}{{\left|Z(0)\right|}^{2}}+\left(1-\varepsilon \right)\frac{\sum_{\tau \neq 0}{\left|Z(\tau )\right|}^{2}}{{\left|Z(\tau )\right|}^{2}}]\;\\ s.t.\;\;\;\;0\leq \varepsilon \leq 1\\ \;\;\;\;\;\;\;\;0\leq \psi_{n}\leq 2\pi \\ \;\;\;\;\;\;\;\;\Delta f/T\leq k_{n}\leq N_{B}\Delta f/T \end{array}$$
it can be seen that ε=1 denotes completely PSL optimization and ε=0 denotes ISL optimization, while 0<ε<1 for both peak sidelobe and integrated sidelobe optimization. The specific processing of the neural networks can be obtained in Algorithm 1.
**Algorithm 1** In the dual-channel neural networks-based waveform optimization method, *N* represents the number of transmitting antennas, ε is a specific optimization factor, and *D* denotes the total number of iterations. Convolution kernel parameters are initialized by truncated random normal distribution data.1:**for** *d* = 1…*D* **do**2:   Inputs are propagated forward; the output phase data $\Psi_{d}=\left[\psi_{1},\ldots ,\psi_{n},\ldots ,\psi_{N}\right]$ and bandwidth data $K_{d}=\left[k_{1},\ldots ,k_{n}\ldots ,k_{N}\right]$ are obtained from the two channels;3:   The MIMO-LFM waveforms can be obtained by $S_{d}=\left[s_{1}\left(t\right),\ldots ,s_{n}\left(t\right),\ldots ,s_{N}\left(t\right)\right]$ and $s_{n}\left(t\right)=a\left(t\right)\exp\left(j\pi k_{n}t^{2}\right)\exp\left(j2\pi f_{n}t\right)\exp\left(\psi_{n}\right)$;4:   The synthesized signal processing result Z(τ) of $S_{d}$ is obtained via (11);5:   Calculate the optimization objective function of Z(τ) described by (22);6:   Back-propagation is employed to optimize the optimization objective function, and the neural networks parameters are updated.7:**end for**8:**return** According to Equation (18), the final phase data $\psi_{D}$ and bandwidth data $K_{D}$ are outputted and the MIMO-LFM waveforms are $S_{D}$.

## 4. Numerical Simulation Analysis

For this section, extensive experiments and numerical analysis were conducted on the effects of optimization performance and the optimization factor, to validate the efficacy of the proposed method. In all the simulations, a MIMO radar with OFDM-LFM waveforms and a half-wavelength uniform linear array containing *N* = 25 elements were considered. The corresponding parameters were Δf = 1 M, a pulse width of $T=\frac{W}{\Delta f}$, a carrier frequency set of $\left[-12,-11,\ldots ,11,12\right]\Delta f$, and a bandwidth modulation factor of $N_{B}=16$. All the simulations were implemented using TensorFlow 1.14.0 on a workstation with GeForce RTX 2080 Ti (NVIDIA, Santa Clara, CA, USA) and 11 G memory.

1. The optimization performance of the proposed method on the PSL and ISL was validated, where we set W = 4, corresponding to a pulse width of T = 4 us. As shown in Figure 4, the original waveforms showed significant periodic high-range sidelobes at ±1, ±2, ±3, and ±4 us, which was consistent with the previous analysis results.

The proposed dual-channel neural networks-based method was applied, to optimize the bandwidth and phase of the OFDM-LFM waveform. Figure 5 shows the results when the optimization factor ε was set to 1, i.e., with PSL as the optimization objective. The initial OFDM-LFM waveforms exhibited periodic high sidelobes after synthesized processing, due to integer multiple relationships between bandwidth and pulse width, as well as the same phase. The initial inputs of the dual-channel neural networks are random phase data and bandwidth data, which can affect the periodicity of the OFDM-LFM sidelobe. In other words, the locations of high-range sidelobes are no longer at $\tau =\pm \frac{m}{\Delta f}$, specifically at the positions of ±1, ±2, ±3, and ±4 us in the Figure 5a. However, this does not improve or even worsen the sidelobe properties. Fortunately, after the optimization by the dual-channel neural networks proposed in this paper, the sidelobe performance of the waveform considerably improved. The PSL values of the initial OFDM-LFM waveforms, the waveforms with random parameters, and the waveforms optimized by the neural networks were −13.888 dB, −12.952 dB, and −25.256 dB, respectively. After optimization with the dual-channel neural networks, the periodicity of the sidelobe position and the PSL performance of the OFDM-LFM waveforms improved. Figure 5b shows the curve of the loss function with the number of iterations during optimization. The loss function rapidly decreased in the first few iterations and tended to converge after 500 iterations, indicating that the proposed method has good convergence.

Subsequently, the optimization objective function was adjusted by modifying the optimization factor ε. When ε = 0, the sidelobe performance was optimized with the ISL as the objective, and the corresponding results were shown in Figure 6a. The ISL values for the initial waveforms, the random parameter waveforms, and the optimized waveforms were −5.259 dB, 1.238 dB, and −6.033 dB, respectively. The iterative results of the loss function during the optimization are shown in Figure 6b, which shows that the decreasing loss function tended to converge after 100 iterations.

The simulation results show that compared with the traditional OFDM-LFM signal, the waveform design method based on dual-channel neural networks proposed in this paper exhibited better performance, in terms of the PSL and ISL. This improved the sidelobe properties, which were no longer periodic, and promoted rapid convergence of the algorithm. The optimized OFDM-LFM waveforms were more conducive to target detection.

2. A new optimization objective function is constructed when optimizing the OFDM-LFM waveforms using neural networks. As described in Equation (Equation 22), the optimization factor ε can be flexibly adjusted between the PSL and ISL, with the PSL as the optimization objective when ε=1 and ε=0 for the ISL. When 0<ε<1, both the PSL and the ISL are optimized. Figure 7 shows the optimization results when W = 6 and other parameters were similar to those in Experiment 1, with ε set as 1, 0.8, 0.5, 0.2, and 0.

It can be seen that as ε gradually increased between 0 and 1, the PSL value of the optimized waveform decreased, i.e., better PSL optimization results were obtained. Simultaneously, the ISL gradually increased, indicating that the ISL of the optimized waveform gradually deteriorated. The minimum value of the PSL was −24.348 dB when ε = 1, whereas its maximum value was −16.499 dB when ε = 0. The minimum and maximum values of the ISL were −4.075 and −2.476 dB, respectively. The corresponding optimization results are shown in Table 1.

## 5. Conclusions

In this paper, a waveform optimization method based on dual-channel CNNs was proposed, to address the problem of high-range sidelobe levels after pulse compression in MIMO radar with OFDM-LFM waveforms. We constructed channels for bandwidth and phase optimization based on ResNeXt and CNN, and we considerably reduced the sidelobe levels of the waveform by the novel phase and bandwidth loss functions. Moreover, an optimization factor was employed, to balance the PSL and ISL performance of the waveforms. Our simulation and analysis results show that the synthesized processing results of the designed waveforms exhibited better correlation properties than the conventional OFDM-LFM waveforms, thereby improving the target detection performance of the radar system. In our ensuing research, under the existing dual-channel neural networks framework we will consider introducing Doppler sensitivity in the MIMO-LFM waveform design, to improve the detection performance of radar systems for high-speed moving targets.

## Figures and Tables

**Figure 1 sensors-24-05471-f001:**
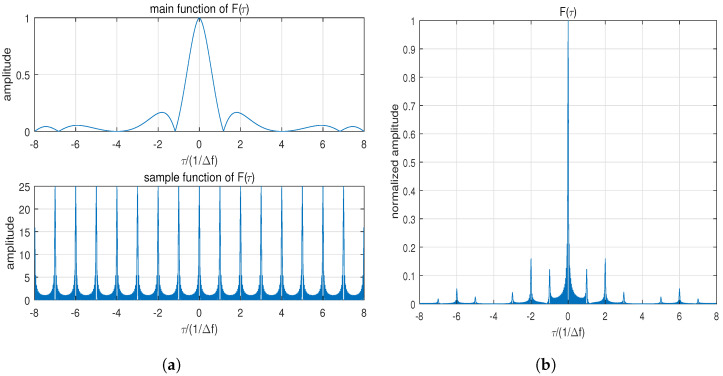
Mathematical analysis of F(τ): (**a**) Main function and periodic sampling function with *N* = 25. (**b**) Correlation result of MIMO-LFM with *N* = 25.

**Figure 2 sensors-24-05471-f002:**
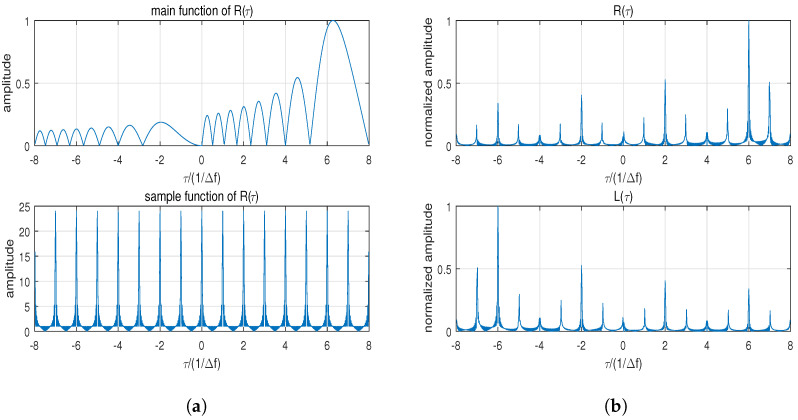
Mathematical analysis of R(τ): (**a**) Main function and periodic sampling function of R(τ). (**b**) Results of R(τ) and L(τ).

**Figure 3 sensors-24-05471-f003:**
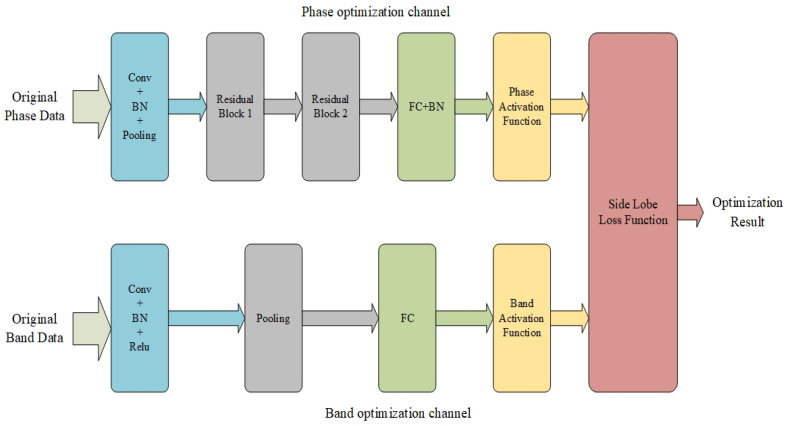
Dual-channel CNNs for OFDM-LFM waveform design. In the figure, “Conv” represents convolution layer; “BN” represents batch normalization; “Pooling” represents max-pooling; “FC” represents fully connected layer; “Residual block” represents residual mapping based on group convolution; “phase activation function” and “bandwidth activation function” are the phase and bandwidth activation functions proposed in this paper for OFDM-LFM waveform design; “side lobe loss function” represents the objective function based on the sidelobe properties optimization represented by Equation (Equation 22).

**Figure 4 sensors-24-05471-f004:**
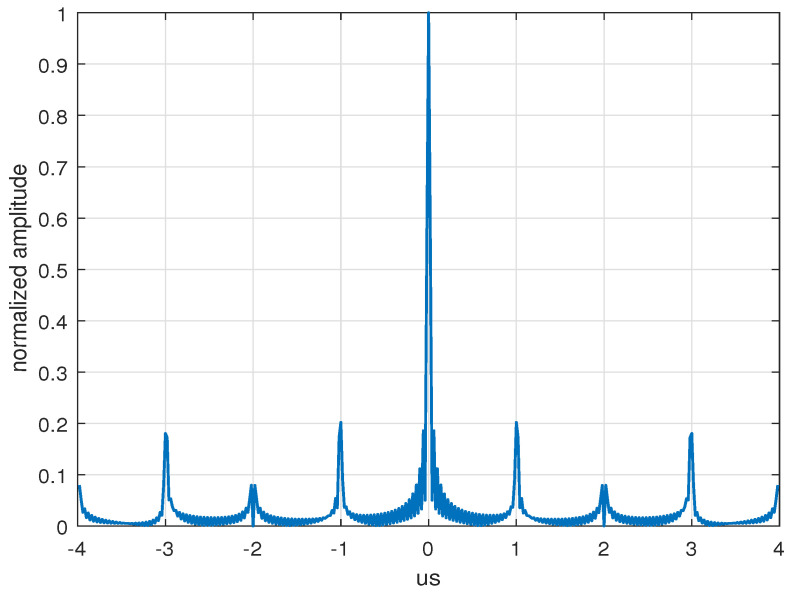
Sidelobe properties of the initial OFDM-LFM waveforms.

**Figure 5 sensors-24-05471-f005:**
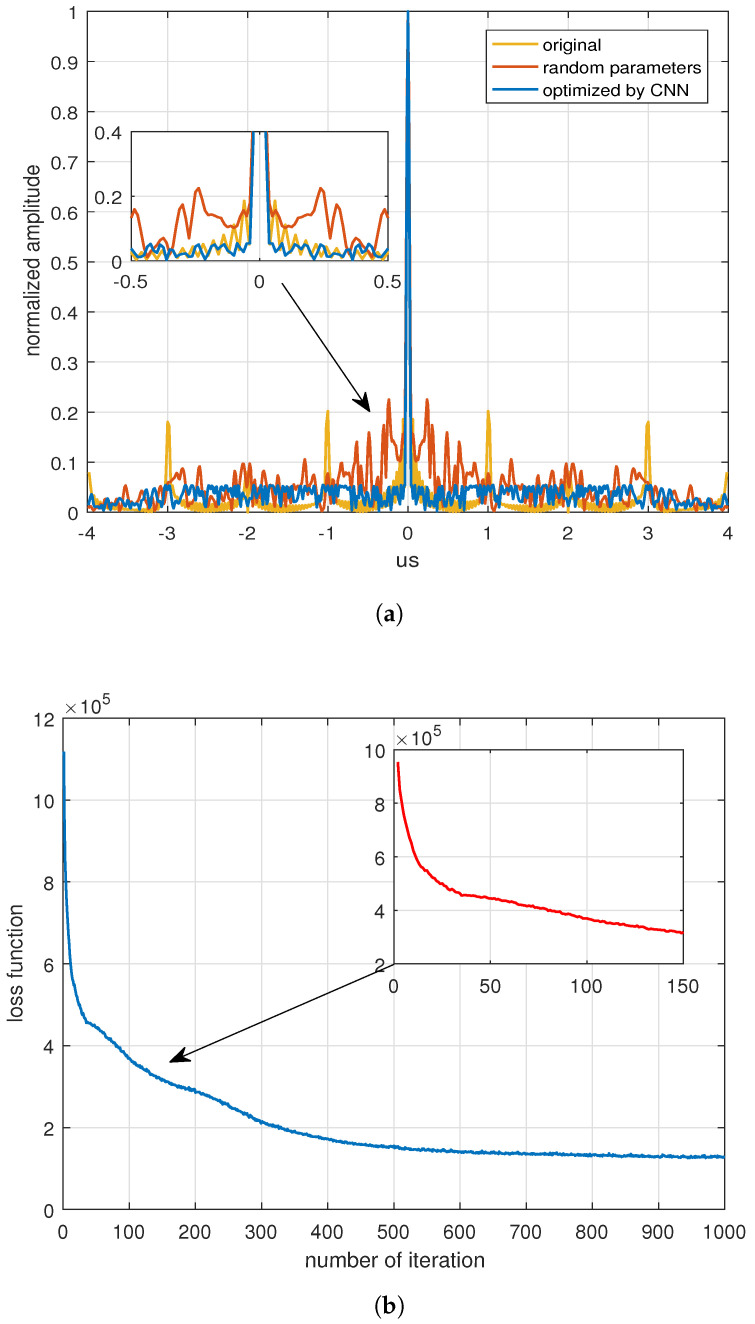
Optimization with the PSL as the objective (ε = 1): (**a**) Optimized results with the PSL. (**b**) The descent of the loss function during iterations with the PSL.

**Figure 6 sensors-24-05471-f006:**
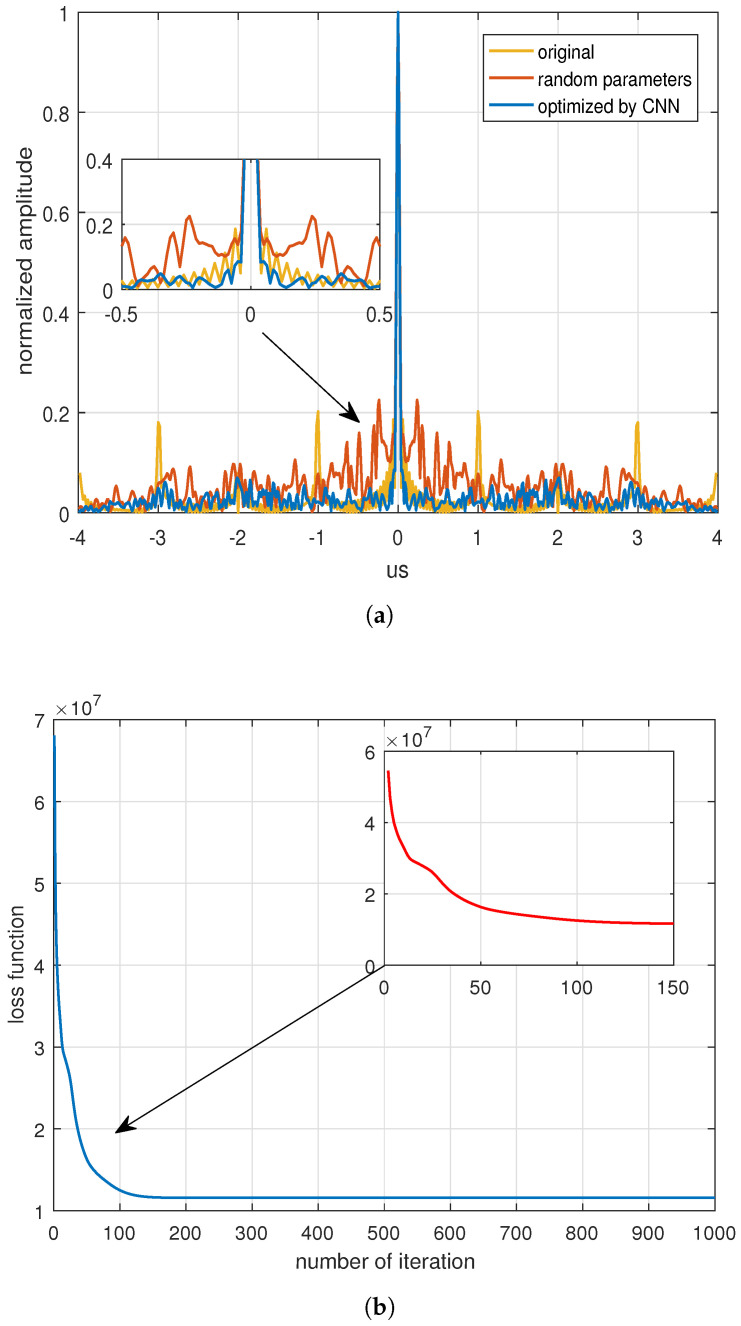
Optimization with the ISL as the objective (ε = 0): (**a**) Optimized results with the ISL. (**b**) The descent of the loss function during iterations with the ISL.

**Figure 7 sensors-24-05471-f007:**
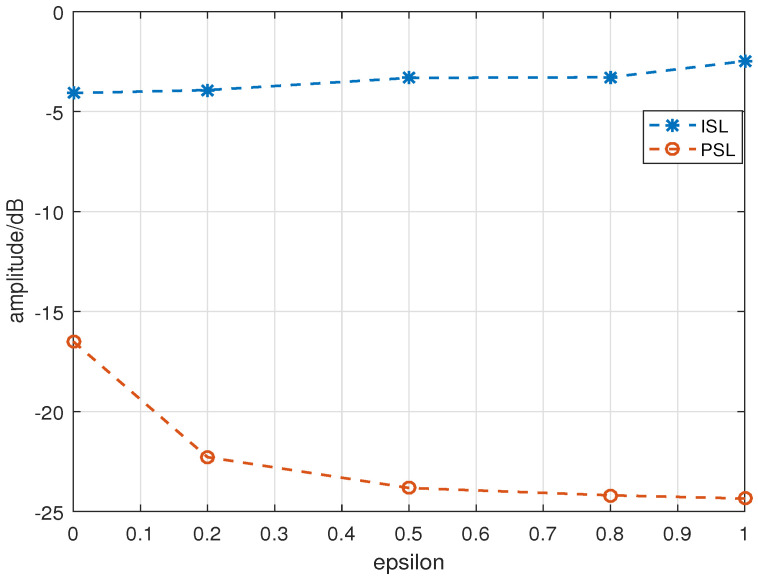
Results of different optimization factors.

**Table 1 sensors-24-05471-t001:** Optimization results with different factors.

ε	0	0.2	0.5	0.8	1.0
ISL	−4.075 dB	−3.926 dB	−3.317 dB	−3.284 dB	−2.476 dB
PSL	−16.499 dB	−22.281 dB	−23.818 dB	−24.186 dB	−24.348 dB

## Data Availability

The data are contained within the article.

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
