# Peer review of "A Novel Waveform Optimization Method for Orthogonal-Frequency Multiple-Input Multiple-Output Radar Based on Dual-Channel Neural Networks"

_sensors, 2024, doi:10.3390/s24175471_

Round 1
Reviewer 1 Report
Comments and Suggestions for Authors
In this paper, the authors proposed a waveform optimization method based on dual-channel CNNs in order to address the problem of high-range sidelobe levels after pulse compression in MIMO radar with OFDM-LFM waveforms.
In order to improve the quality of the article, the authors must consider the following aspects:
- In sections 2 and 3 there is not any citation. Must be clearly presented which of the formulas are from literature and which are the results of the work of the authors
- - To follow much easier the mathematical results an appendix is needed with the demonstrations of the obtained formulas
- - In section 3 is presented Algorithm 1. Because there is not another algorithm in the paper, this should be named simply The Algorithm
- - In section 4 are presented the experiments and numerical analysis only for N=25 elements. The authors should perform experiments also for other values in order to show that their optimization method is good
- - Also, the Conclusions section must be improved. the authors must emphasize more clearly what is new in this article compared to other works in the field
Reviewer 2 Report
Comments and Suggestions for Authors
This paper proposes a novel waveform optimization method for MIMO radar systems utilizing OFDM-LFM waveforms. The method leverages dual-channel neural networks to optimize the phase and bandwidth of the waveforms, aiming to reduce the peak sidelobe levels (PSL) and integral sidelobe levels (ISL). The paper includes theoretical analysis, numerical simulations, and comparisons with traditional methods.
1:The authors introduce a dual-channel CNN approach to optimize OFDM-LFM waveforms, which is innovative. However, the paper should more clearly differentiate its contributions from existing works that also use deep learning for radar waveform optimization. Explicitly stating the novelty in the introduction would strengthen the manuscript.
2:The simulation results are promising, showing a significant reduction in PSL and ISL. However, the paper would benefit from a more extensive comparison with other state-of-the-art methods beyond the traditional OFDM-LFM. Including methods such as those based on blind source separation or deep learning in MIMO radar, would provide a stronger validation of the proposed method's effectiveness.
3:The introduction of the optimization factor ε to balance PSL and ISL is interesting. However, its impact on practical radar performance (e.g., target detection accuracy) should be discussed. Including a sensitivity analysis showing how different ε values affect overall radar performance would be beneficial.
4:Deep learning models, especially CNNs, can be computationally intensive. A discussion on the computational requirements and efficiency of the proposed method compared to traditional optimization techniques would be valuable. This could include training time, inference speed, and resource requirements.
5:Suggested Additional References:
An Advanced Scheme for Range Ambiguity Suppression of Spaceborne SAR Based on Blind Source Separation
This reference could provide insights into advanced signal processing techniques that could complement the proposed method, particularly in handling range ambiguities which might be relevant in MIMO radar contexts.
MIMO Radar Imaging Method with Non-Orthogonal Waveforms Based on Deep Learning
This reference could provide a comparative framework for the deep learning-based approach in MIMO radar waveform design, offering a basis for a more robust discussion on the advantages and potential limitations of the proposed method.
6:A critical technical issue is that reducing sidelobes often leads to pulse broadening, which reduces resolution. The paper does not address this trade-off, which is crucial for the practical application of the method. The authors must consider and discuss the impact of their optimization on resolution, as merely reducing sidelobes without accounting for resolution is insufficient.
Comments on the Quality of English Language
1:Ensure all equations are properly numbered and referenced in the text. For example, Equation (18) should be referenced explicitly when discussing the optimized waveform formulation.
2:The quality and readability of figures could be improved. Ensure all text within figures is legible and color choices provide clear contrast.
3:There are minor grammatical errors and awkward phrasings throughout the manuscript.
4:Reference formatting is inconsistent. Ensure all references follow the journal's style guide. For example, ensure proper capitalization in titles and consistent formatting for journal names and volumes.
Round 2
Reviewer 2 Report
Comments and Suggestions for Authors
Accept in present form